# Acetylcholinesterase electrochemical biosensors with graphene-transition metal carbides nanocomposites modified for detection of organophosphate pesticides

Bo Wang[1,2], Yiru Li[2], Huaying Hu[2], Wenhao Shu[2], Lianqiao Yang[2]*, Jianhua Zhang[2]

1 Microelectronics Research & Develop Center, Shanghai University, Shanghai, China, 2 Key Laboratory of Advanced Display and System Applications, Ministry of Education, Shanghai University, Shanghai, China

* yanglianqiao@i.shu.edu.cn

**Data Availability Statement:** All relevant data are within the paper and its Supporting Information files.

## Abstract

An acetylcholinesterase biosensor modified with graphene and transition metal carbides was prepared to detect organophosphorus pesticides. Cyclic voltammetry, differential pulse voltammetry, and electrochemical impedance spectroscopy were used to characterize the electrochemical catalysis of the biosensor: acetylcholinesterase/chitosan-transition metal carbides/graphene/glassy carbon electrode. With the joint modification of graphene and transition metal carbides, the biosensor has a good performance in detecting dichlorvos with a linear relationship from 11.31 μM to 22.6 nM and the limit of detection was 14.45 nM. Under the premise of parameter optimization, the biosensor showed a good catalytic performance for acetylcholine. Compared to the biosensors without modification, it expressed a better catalytic performance due to the excellent electrical properties, biocompatibility and high specific surface area of graphene, transition metal carbides. Finally, the biosensor exhibits good stability, which can be stored at room temperature for one month without significant performance degradation, and has practical potential for sample testing.

## Introduction

With the continuous development of the global economy and population, the demand for foods such as vegetables and fruits are increasing. In order to meet the growing demand for food, especially vegetables, and resolve the problem of insects living with the crop, the using of pesticides in agriculture has become widespread in the past few decades. Among them, organophosphorus pesticides (OPs) were widely used because of their fast response and low cost. However, humans and livestock were posed life-threatening risks by OPs because OPs would inhibit the catalytic activity of acetylcholinesterase (AChE), causing acetylcholine (ATCl) to accumulate in the body without hydrolysis in time, causing damage to the nervous system. In view of these, it is necessary to detect the residual of OPs in food [1–5].

In the past few decades, the method of detecting OPs has been developed a lot and there were already mature technology and applications in liquid chromatography, mass spectrometry and so on [6, 7]. However, timely, rapid and easy detection of OPs was still a big challenge because traditional methods require professional equipment and person [8, 9].

**Funding:** National Natural Science Foundation of China (51505270).

**Competing interests:** The authors have declared that no competing interests exist.

Electrochemical methods are receiving more and more attention because of the advantages of simple instruments, fast result acquirement, high reliability, easiness for operation, high sensitivity and compatible with complex samples. ATCl can be catalyzed by AChE to produce thiocholine (TCl), and the electron loss associated with the irreversible oxidation of TCl could be detected by electrochemical workstation. In the presence of OPs, the amount of TCl would be reduced due to the inhibition of AChE. According to this mechanism, the OPs residue of the analyte can be easily detected by the electrochemical biosensor [10–14].

There are two key points in the preparation of electrochemical biosensors. One is keeping the catalytic activity of the enzyme after immobilized on the electrode. Another is the issue of electrochemical signal transmission associated with oxidation of TCl, because of the non-conductivity of the biological enzyme. Chitosan (CS) was a non-toxic natural hydrophilic polysaccharide with good biocompatibility, adhesion and excellent film-forming ability. It has been widely used in electrode modification and enzyme immobilization [15–17]. Recently, $Ti_3C_2T_x$ (T represents the terminating groups, x represents the number of these terminating groups), one member of transition metal carbides (MXenes) family, a new two-dimensional nanomaterial was discovered [18–21]. MXenes has been tried and applied in many fields such as bio- and gas-sensors, energy storage, electromagnetic interference shielding, reinforcement for composites, water purification, lubrication, and chemical, photo- and electro-catalysis due to their many advantages, such as large surface area, good hydrophilicity, and so on [22, 23]. In the application of electrochemical biosensors, MXenes has achieved satisfactory results. Quan et al. used 3D sodium titanate nanoribbons synthesized by MXene to modify electrodes for use in electrochemical biosensors to detect PSA [24]. Haiyuan et al. made DNA probes on MXene, prepared electrochemical immunosensors, and used them to detect the breast cancer marker Mucin1 [25].Wang et al. developed an electrochemical biosensor based on DNA nanostructures and MXene to detect mycotoxins and the biosensor could achieve a low limit of detection of 5 pM in the range of 5 pM to10 nM [26].

Compared with our previous work, graphene (GR) and silver nanowire composite nanofilms were used as the amplification strategy, and $TiO_2$ film was used as the immobilization matrix [27]. Although the biosensors exhibit good performance, the fabrication process and the materials required are complex. In order to enhance the application potential, the preparation process and the required materials of biosensor should be simplified and its electrocatalytic performance cannot be affected.

In this article, dichlorvos (DDVP) was used as the representative of OPs, and GR was used as the electrode modification material, $Ti_3C_2T_x$-CS was used as the immobilization matrix of the enzyme and an electrochemical biosensor was fabricated with structure of AChE/$Ti_3C_2T_x$-CS/GR/glassy carbon electrode (GCE)(Fig 1). Here, $Ti_3C_2T_x$ has two functions. First of all, $Ti_3C_2T_x$ is a hydrophilic, biocompatible nanomaterial, which can effectively improve the efficiency of enzyme fixation [23]. Secondly, $Ti_3C_2T_x$ has a large specific surface area and conductivity, which can improve the electrical performance of the electrode [28]. The biosensor manufacturing process was optimized to improve performance. The catalytic ability, DDVP detection and stability were characterized by biosensors. Finally, the biosensor was tested on the real sample with tap water to verify its applicability.

## Experimental

### Materials and chemicals

AChE (from electric eel), ATCl, and DDVP were purchased from Sigma-Aldrich. CS (viscosity >400 mPa s), Bovine serum albumin (BSA) and GR (stripped by 1-Methyl-2-pyrrolidinone (NMP)) were from Aladdin Bio-Chem Technology (Shanghai, China). $Ti_3C_2T_x$ was from

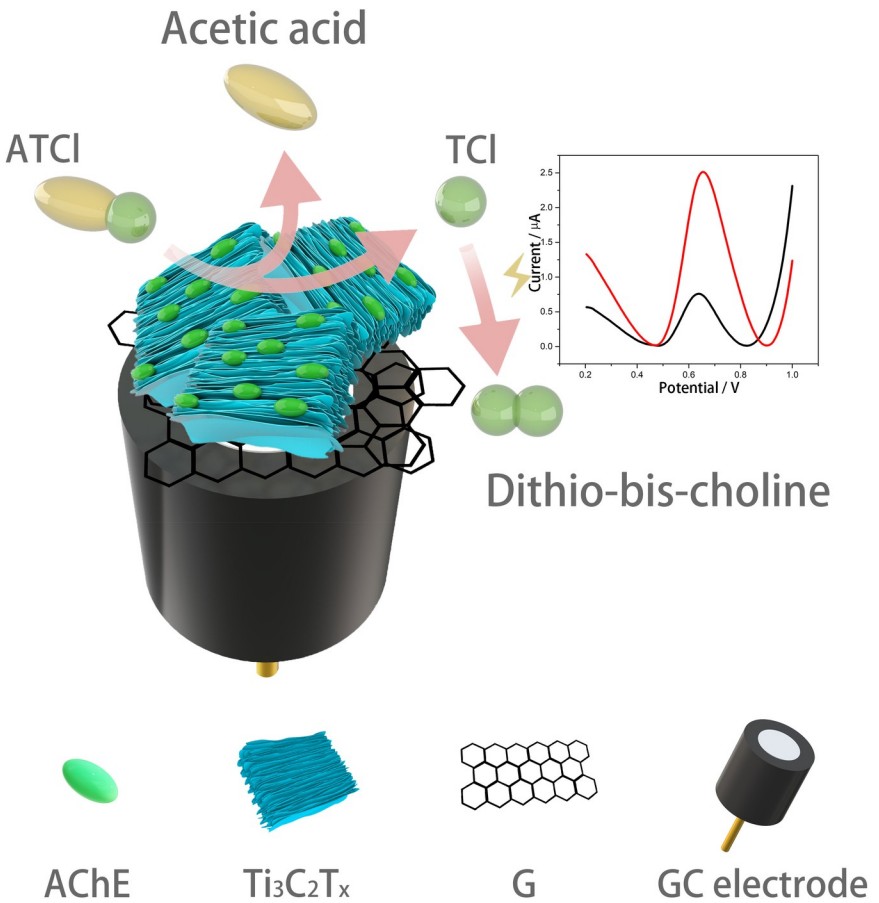

**Fig 1. The structure of the AChE biosensor and the principle of detecting organophosphorus pesticides.**

HAOXI Research Nanomaterials. The other chemicals used in the experiments were from Sinopharm Chemical Reagent Co., Ltd (Shanghai, China). All chemical reagents used are of analytical grade. Nitrogen with a purity of 99.9% comes from Air Liquide (China) Holding Co., Ltd.

## Apparatus and measurement

Hitachi Regulus 8220-SEM, FEI Tecnai G2 F20 (USA), ThermoFisher 250Xi was used for scanning electron microscope (SEM), transmission electron microscope (TEM), X-ray photoelectron spectroscopy (XPS) characterization of samples. All electrochemical test equipment comes from Shanghai Chenhua Instrument (China), including CHI660E electrochemical workstation, GC working electrode (3 mm), Ag/AgCl/KCl (3 M) reference electrode and platinum wire counter electrode.

## Preparation of GR, $Ti_3C_2T_x$ suspension and CS solution

GR: The GR solution was obtained by GR peeled off with NMP intercalation. It was diluted to 0.4 mg/ml with DMF, and finally sonicate for 30 minutes, seal and store for use.

$Ti_3C_2T_x$: The $Ti_3AlC_2$ powder was dispersed in HF solution, and after centrifugation, stirring, separation, washing and drying, a solid $Ti_3C_2T_x$ powder was obtained. When using, it was ultrasonically dispersed in the 0.2% CS solution to obtain $Ti_3C_2T_x$-CS solution (0.25 mg/ml), ready for use.

CS: The 1% CS solution was prepared by slowly disssolving 1 g of CS in 100 ml of 1% (vol %) glacial acetic acid with stirring to obtain a clear transparent solution, and then diluting the 1% CS solution to various concentrations with DI water.

## Fabrication of the biosensor

The GCE was cleaned through polishing to a mirror surface with 0.3, 0.06 μm alumina powder, cleaning in an ultrasonic bath and blown dry with nitrogen. Then, 4 μL of GR, $Ti_3C_2T_x$-CS solution was dropped on the surface of the GCE, and after completely drying in air, 4 μL of AChE (5 mg/ml in 1% BSA) solution was dropped and dried overnight.

## Electrocatalytic and sensing performance

The sensing properties of the modified electrode in the experiment were mainly characterized by electrochemical methods. Among them, cyclic voltammetry (CV) and electrochemical impedance spectroscopy (EIS) were characterized in 5 mM $K_3[Fe(CN)_6]$ supported by 1 M KCl and 0.1 M KCl containing equimolar $[Fe(CN)_6]^{3-/4-}$ (10/10 mM), respectively. The catalytic analysis of the biosensor was performed using differential pulse analysis (DPV) in 1 mM ATCl, which voltage range was from 0.2 to 1.0 V; amplitude, 0.05 V; pulse width, 0.005 s; pulse period, 0.02 s.

Prior to detection of ATCl and DDVP, the prepared biosensor was tested CV multiple times in phosphate buffered saline (PBS) until a stable curve appeared. When detecting ATCl, the sensor was incubated in ATCl for 5 min and then tested with DPV. And when DDVP was detected, the sensor was incubated in DDVP for 3 min, washed with PBS, and placed in 1 mM ATCl for 5 min to finally test DPV. The inhibition rate (Inhbit%) of the DDVP to the sensor is calculated using Eq (1) [29].

$$\text{Inhibit\%} = 1 - \frac{I_{cat}{}'}{I_{cat}} \tag{1}$$

Where $I_{cat}'$ and $I_{cat}$ were DPV peak currents that were incubated and not incubated in DDVP.

## Results and discussion

### Characterization of GR, $Ti_3C_2T_x$ and AChE

The morphology of the electrode modification material GR, $Ti_3C_2T_x$ and AChE was characterized by SEM and TEM. In Fig 2A, it was obvious that the GR nanosheets have a unique sheet morphology, which was possibly due to the production process producing many small fragments. It can be clearly seen that the multilayer structure of $Ti_3C_2T_x$ and the diameter of a single $Ti_3C_2T_x$ particle was about 13 μm in Fig 2B. From the TEM image, the same multilayer structure can be seen, and the gap between the layers was about 0.8 nm. In the Fig 2C, it was the obvious ice-shaped crystal of AChE.

The composition of $Ti_3C_2T_x$ was probed by XPS analysis. The spectrums of XPS analysis were shown in Fig 3. The detailed information such as element percentage, component name, and component percentage were shown in Table 1, and the analysis results are consistent with previous researches [30]. From the information in Table 1, Al is still present in the prepared $Ti_3C_2T_x$, indicating that the etching is incomplete. Simultaneously, the component at 459.6, 529.8 and 530.5 eV indicate that a part of titanium is oxidized. 31.93% of carbon exists in the form of C-C, which may be formed during the preparation of $Ti_3C_2T_x$ [31].

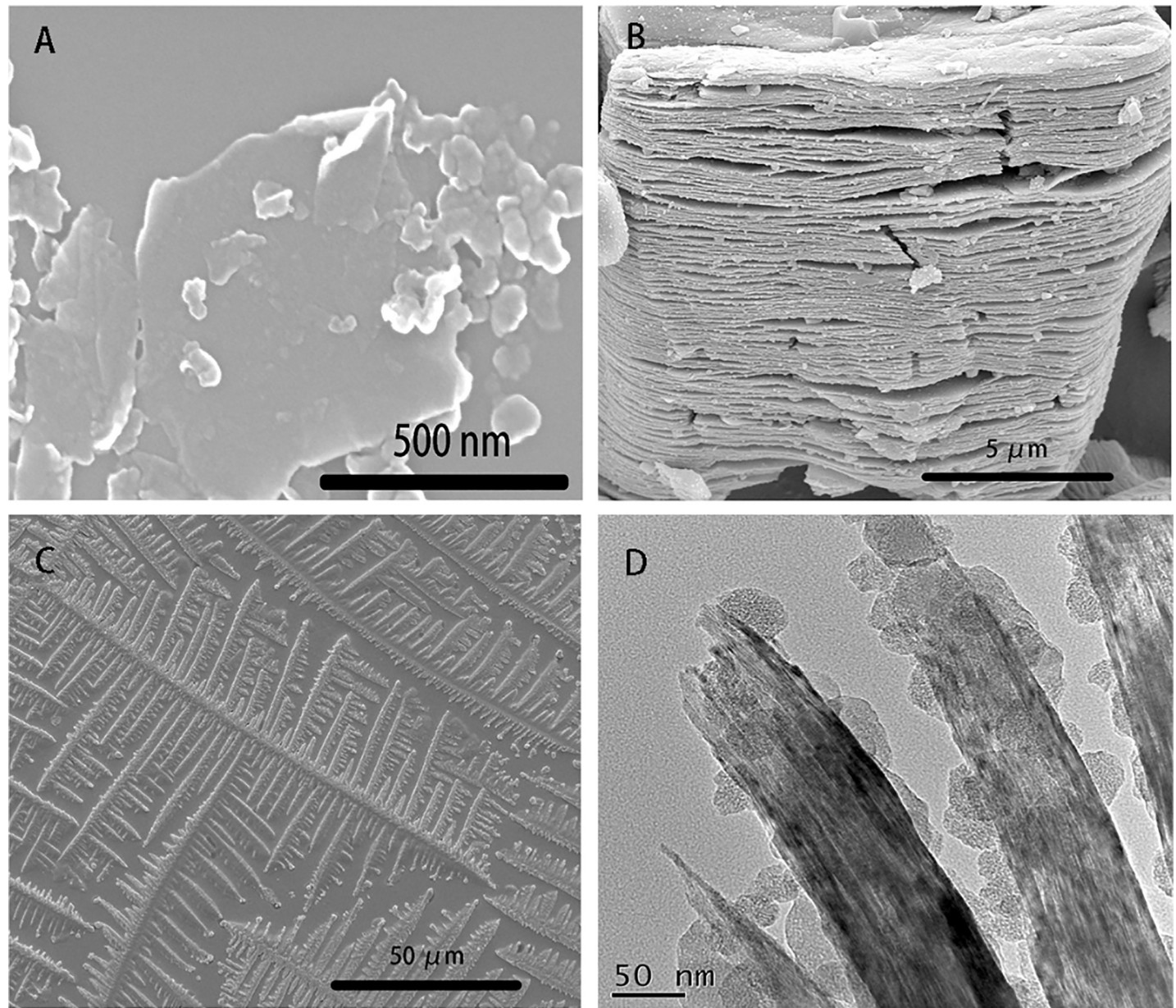

**Fig 2.** The SEM image of (A) GR, (B) $Ti_3C_2T_x$ and (C) AChE, and TEM image of (D) $Ti_3C_2T_x$.

The comparison of MXene before and after etching was probed by XRD analysis shown in Fig 4. After HF etched, the strongest diffraction peak of $Ti_3AlC_2$ in 38.84° (104) disappeared, indicating that the structure of the MAX phase ($Ti_3AlC_2$) material was completely destroyed, and most of the Al layer was etched. In addition, peaks in 9.53° (002) and 19.13° (004) shifted to small angles of 8.9° and 18.31° after the etching. Respectively, their full width at half maximum (FWHM) were larger after the HF acid etching, which should be attributed to the partial replacement of Al by OH-/F- [32, 33]. The peaks at 35.98 (103) and 41.75 (105) are still obvious, indicating that the MAX phase still exists and the etching is incomplete [34].

The electrochemical properties of the modified electrode were investigated by $K_3[Fe(CN)_6]$. In Fig 5A, after the GR modification, the peak current of the CV curve of the electrode was

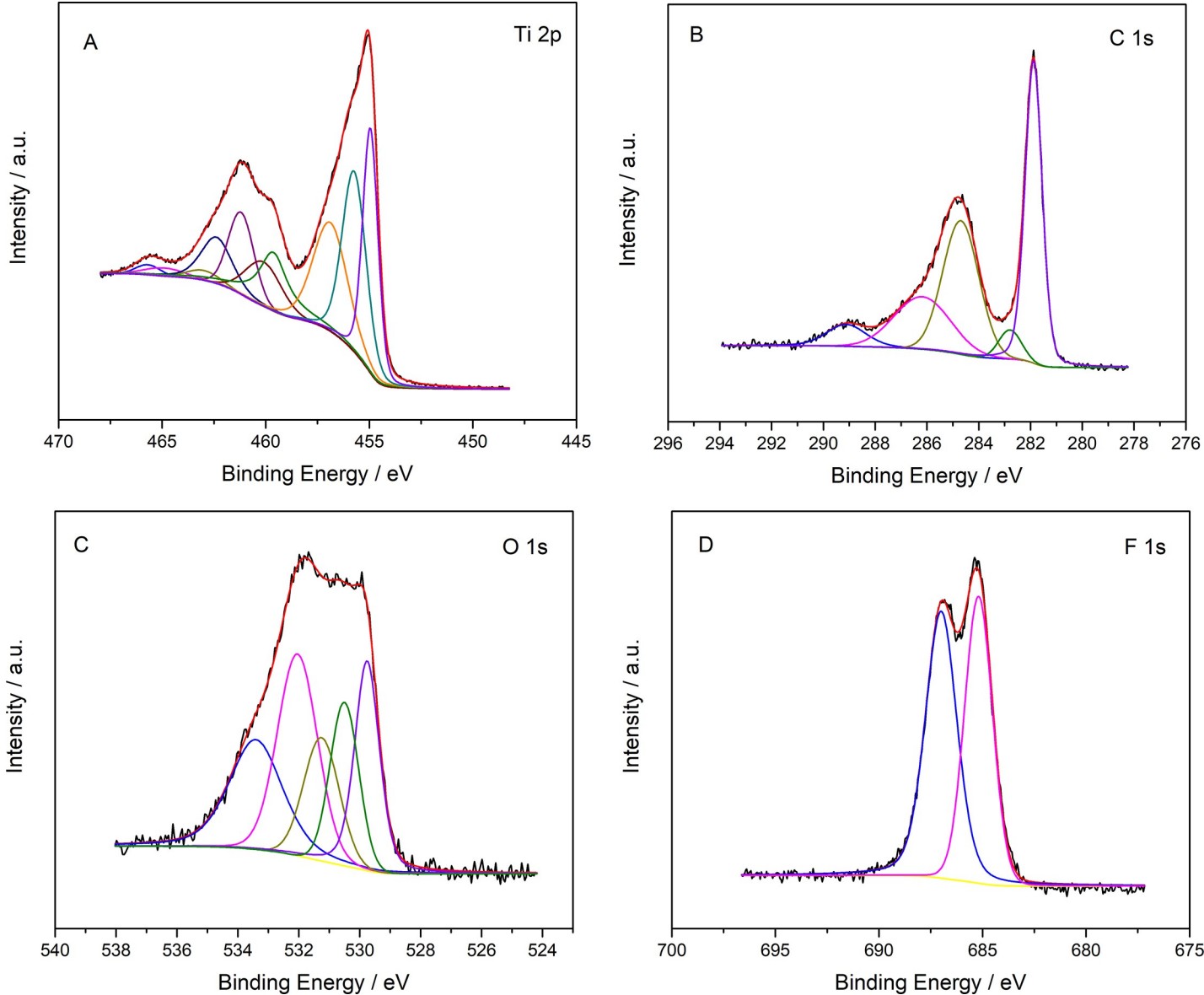

**Fig 3. XPS spectra of crumpled Ti$_3$C$_2$T$_x$.** (A) Ti 2p (B) C 1s (C) O 1s (D) F 1s. Binding energy values of each bond associated with deconvoluted peaks are listed in Table 1.

significantly improved because of the excellent electrical properties of GR. After the addition of Ti$_3$C$_2$T$_x$, the peak current has been also significantly improved, which proves that Ti$_3$C$_2$T$_x$ has good electrical properties. Further, after the addition of Ti$_3$C$_2$T$_x$, the effective surface area of the electrode is improved. According to Eq (2) [35] and Fig 6, the effective surface area of the electrode can be calculated.

$$I_P = 2.69 \times 10^5 n^{3/2} A D_0^{1/2} C_0 \nu^{1/2} \tag{2}$$

Where I$_P$, n, A, D$_0$, C$_0$ and $\nu$ represent the redox peak current (amperes), the electrons oxidized or reduced per molecule, the effective surface area of the electrode (cm$^2$), diffusion coefficient of K$_3$[Fe(CN)$_6$] in 1M KCl ($0.76 \times 10^{-5}$ cm$^2$s$^{-1}$), concentration of redox species

**Table 1. XPS peak fitting results for crumpled $Ti_3C_2T_x$.**

| Element | Overall atomic% | Component name | Component atomic% | BE (eV) | FWHM (eV) |
|---|---|---|---|---|---|
| Ti $2p_{3/2}$ ($2p_{1/2}$) | 22.22 | Ti-C | 24.32 | 454.9 (461.2) | 0.87 (1.45) |
| | | $Ti^{2+}$ | 27.32 | 455.7 (462.4) | 1.36 (1.76) |
| | | $Ti^{3+}$ | 24.16 | 456.9 (463) | 1.95 (1.86) |
| | | $TiO_2$ | 14.75 | 459.6 (464.9) | 1.64 (2.23) |
| | | C-Ti-$F_x$ | 9.45 | 460.1 (465.7) | 2.06 (1.36) |
| C 1s | 29.87 | C-Ti-$T_x$[a] | 42.14 | 281.9 | 0.78 |
| | | C-Ti-$T_x$[a] | | 282.8 | 1.02 |
| | | C-C | 31.93 | 284.7 | 1.64 |
| | | $CH_x$/CO | 19.56 | 286.2 | 2.55 |
| | | COO | 6.37 | 289.1 | 1.79 |
| O 1s | 19.19 | $TiO_2$ | 19.70 | 529.8 | 0.96 |
| | | C-Ti-$O_x$ | 15.48 | 530.5 | 1.11 |
| | | C-Ti-$(OH)_x$ | 14.55 | 531.2 | 1.38 |
| | | $Al_2O_3$ | 27.55 | 532.0 | 1.55 |
| | | $H_2O$[b] | 22.72 | 533.4 | 2.06 |
| F 1s | 28.72 | C-Ti-$F_x$ | 46.32 | 685.2 | 1.59 |
| | | $AlF_x$ | 53.68 | 687.0 | 1.8 |

(mol $cm^{-3}$) and scanning rate (V $s^{-1}$). In Fig 6, as the scan rate decreases, $I_{pa}$ and $I_{pc}$ also decrease and linear with the square root of the scan rate, whether it is a bare electrode or a modified electrode with $Ti_3C_2T_x$-CS and GR. Finally, the effective surface area of the electrode can be calculated from 0.0517 $cm^2$ to 0.0639 $cm^2$ because of the modification of $Ti_3C_2T_x$-CS and GR. Immediately after the addition of AChE to the electrode (Fig 5A), the $I_p$ of the electrode was significantly reduced due to the non-conductivity of the enzyme as a protein.

Similarly, in Fig 5B, EIS was also used to characterize the electrochemical properties of the various layers of the electrode. The EIS technique is used to detect the impedance of the modified electrode during the preparation process. The figure of the Nyquist of the electrode on the complex plane presents a semicircle in high-frequency domain and a straight line in low-frequency domain. Wherein the diameter of the semicircle is related to $R_{CT}$ transfer impedance, whereas the straight portion is related to the diffusion processes. The $R_{CT}$ can be calculated by Eq (3) [36].

$$Z(\omega) = R_s + \frac{R_{CT} + \sigma\omega^{-1/2}}{(C_d\sigma\omega^{1/2} + 1)^2 + \omega^2 C_d^2 (R_{CT} + \sigma\omega^{-1/2})^2} - j\frac{\omega C_d (R_{CT} + \sigma^{-1/2})^2 + \sigma\omega^{-1/2}(\sigma\omega^{1/2}C_d + 1)}{(C_d\sigma\omega^{1/2} + 1)^2 + \omega^2 C_d^2 (R_{CT} + \sigma\omega^{-1/2})^2} \quad (3)$$

Where $R_s$ is the solution resistance, $C_d$ is the double layer capacitance, and $\sigma = \frac{RT}{\sqrt{2}F^2A}\left(\frac{1}{\sqrt{D_0}C_0^*} + \frac{1}{\sqrt{D_R}C_R^*}\right)$, simply [27].

So, the change in $R_{CT}$ of the electrode after adding various modifiers can be calculated that the value of $R_{CT}$ was about 61, 52 and 55 $\Omega$, while the $R_{CT}$ was 73 $\Omega$ of bare electrode after dropped GR, $Ti_3C_2T_x$-CS and AChE on electrode. Similar to CV, the impedance decreases sequentially after the addition of GR and $Ti_3C_2T_x$-CS due to its excellent electrical properties. And the increase in impedance after the addition of AChE was due to its non-conductivity.

## Electrochemical characterization of fabricated AChE biosensors

To further demonstrate the effect of different modifying materials on biosensor performance, the DPV curve with different modifications was employed in 1 mM ATCl as shown in Fig 7. It

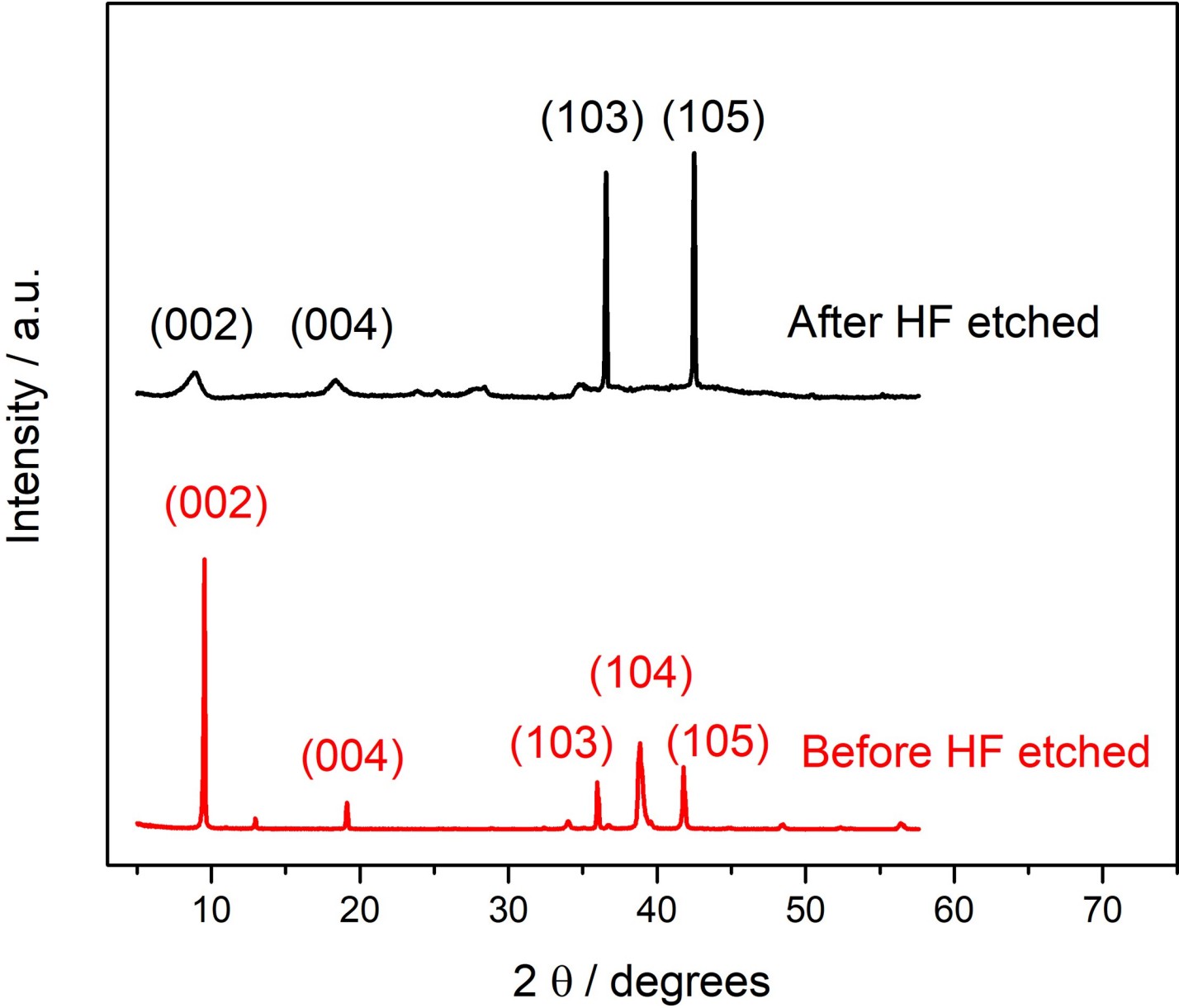

**Fig 4. XRD spectrum of MXene before and after etching with HF.**

is obvious that the curve c with the structure of $AChE/Ti_3C_2T_x$-CS/GR/GCE has the highest peak. Without GR or $Ti_3C_2T_x$, it has a significant reduction in peaks. Among them, when there is no GR, the sensor has the weakest catalytic ability (curve b). This shows that in the sensor, GR acts as the most significant electrochemical signal amplification. Only $Ti_3C_2T_x$ modified biosensors cannot effectively collect electrical signals due to the weak conductivity of CS. When GR and $Ti_3C_2T_x$ are used together, complementary advantages of the electric signal amplification of GR and the hydrophilicity of $Ti_3C_2T_x$ which is easy to immobilize enzymes make the biosensors have better performance [23, 28].

Furthermore, a variety of different concentrations of ATCl were used in DPV testing in order to more fully explain the electrochemical catalysis of the biosensor. As shown in Fig 8, as

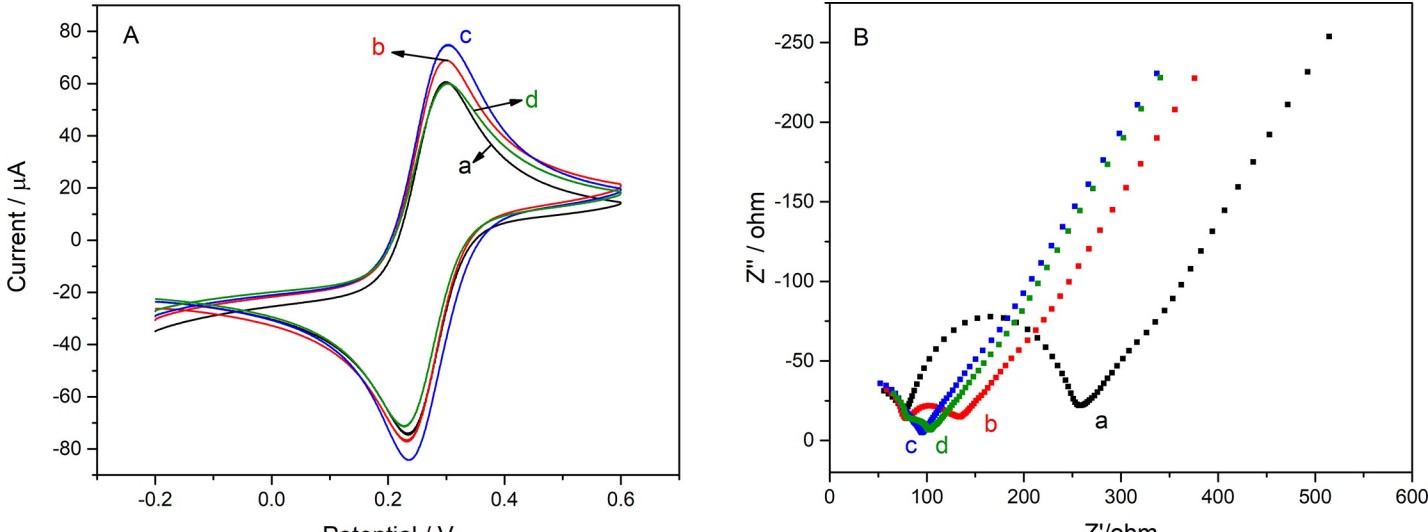

**Fig 5.** (A) CV and (B) EIS characterization of the modified electrode. Among them, the curves and the lattices are (a) bare GC, (b) GR/GC, (c) $Ti_3C_2T_x$-CS/GR/GC and (d) AChE/$Ti_3C_2T_x$-CS/GR/GCE, respectively.

the concentration of ATCl increases, the DPV peak of the sensor was significantly improved. Moreover, the reciprocal of the peak current ($I_{cat}^{-1}$) increased linearly with the reciprocal of the ATCl concentration ($C_{ATCl}^{-1}$) increased: $I_{cat}^{-1} = 0.34877 C_{ATCl}^{-1} - 0.07384$ ($R^2 = 0.9944$). The value of the apparent Michaelis–Menten constant ($K_m$) was calculated to be 4.89 mM, according to Lineweaver-Burk equation (Eq (3)) [37].

$$\frac{1}{I_{cat}} = \frac{K_m}{I_{max}} \times \frac{1}{C_{ATCl}} + \frac{1}{I_{max}} \qquad (4)$$

Where $I_{max}$ was the maximum current measured under saturated substrate condition.

## Optimization of the biosensor

The fabrication process parameters of the biosensor were optimized. Fig 9 shows the statistic results of the $I_{cat}$ values in response to 1 mM ATCl for different pH values and enzyme loadings. Five pH values: 6.5, 7.0, 7.5, 8.0, 8.5 of 1 mM ATCl solution were prepared to detect the catalytic results of the biosensor. The experiments were repeated three times while the other variables remained unchanged. It was apparent that as the pH values increases, the value of $I_{cat}$ increases first then decreases and shows the largest $I_{cat}$ values at pH 7.5. Therefore 7.5 was used as the most suitable pH value during the testing of the biosensor. Similarly, when the other parameters keeping unchanged and the enzyme loading was 2, 3, 4, 5, 6 μL, the $I_{cat}$ value of the biosensor increases in first and then decreases, and reaches the maximum when the enzyme loading is 4 μL. Therefore, 4 μL was chosen as the best enzyme loading values.

## Detection of the pesticide and real sample

The relationship between Inhibit% and pesticide concentration was studied. The AChE electrochemical biosensor was immersed in different concentrations of DDVP solution for 3 min. Subsequently, the sensor's Inhibit% of ATCl catalytic ability was detected, calculated and shown in Fig 10. The entire test process was repeated three times. From the Fig 10, it can see that the Inhibit% was linear with respect to the logarithm of DDVP concentration: Inhibit% =

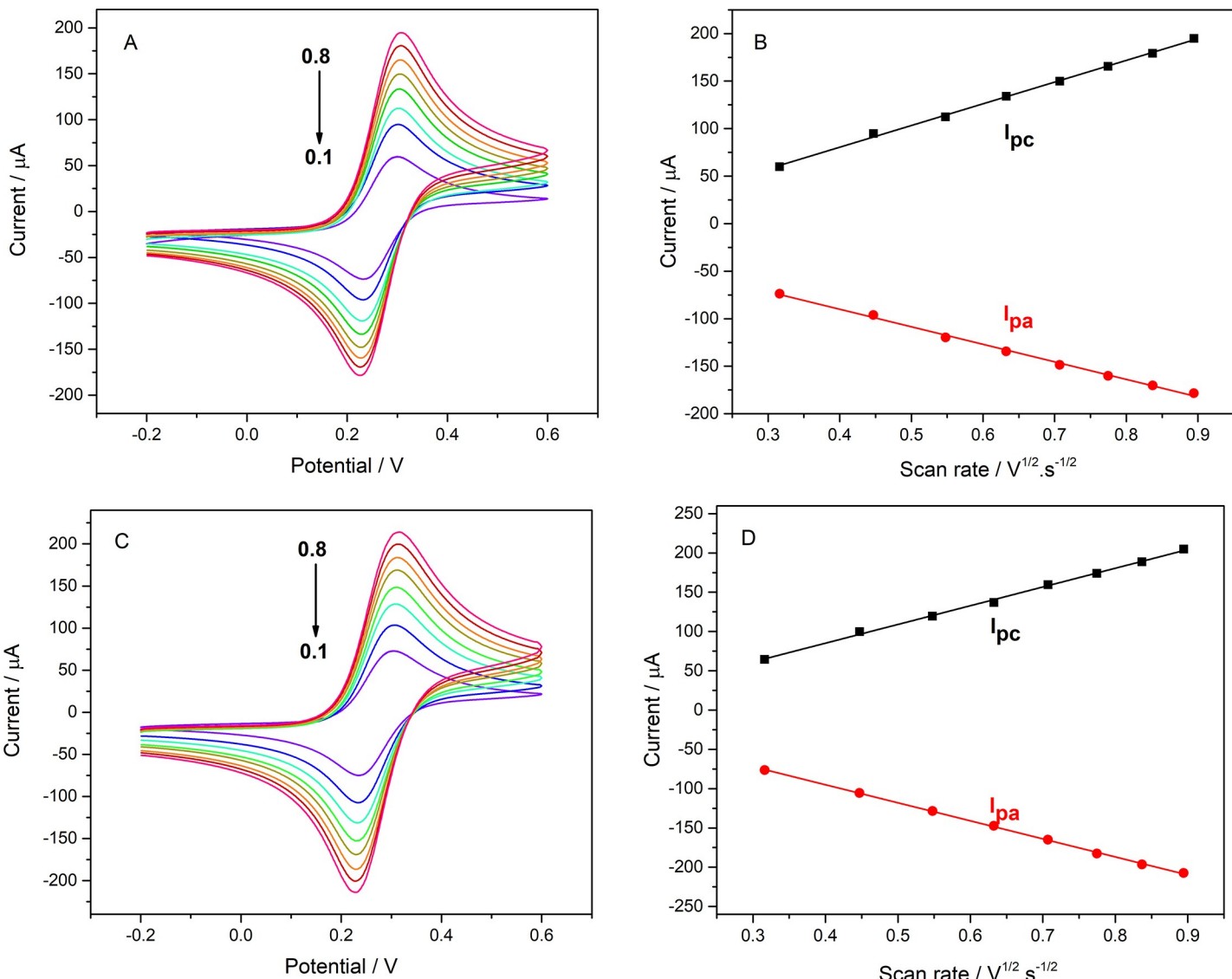

**Fig 6.** CVs of (A) bare GCE and (C) $Ti_3C_2T_x$-CS/GR/GCE with different scan rate (from 0.1~0.8 Vs$^{-1}$). And the anodic peak current ($I_{pa}$) and the cathodic peak current ($I_{pc}$) vs. the square root of the scan rate in the CV curves of the (B) GCE, and the (D) $Ti_3C_2T_x$-CS/GR/GCE.

14.64923lgC$_{DDVP}$+123.085 (R$^2$ = 0.9977) with the limit of detection (LOD) was 14.45 nM (3.2 μg/L) (calculated in a 3σ rule). And the value of LOD was much lower than the allowable concentration of DDVP in the centralized domestic water surface water source standard in China's surface water environmental quality standard (GB3838-2002).

In order to verify the performance of the sensor in practical applications, the urban tap water was used as the real sample to characterize the biosensor as shown in Table 2. Three different concentrations of DDVP real samples were tested and the Recovery% were 98.2%, 109% and 99%. The results show that the biosensor has good practical application ability.

## Repeatability, stability and selectivity

Moreover, repeatability, stability, and selectivity are some of the most important indicators of sensor performance. Five biosensors were simultaneously fabricated to characterize the

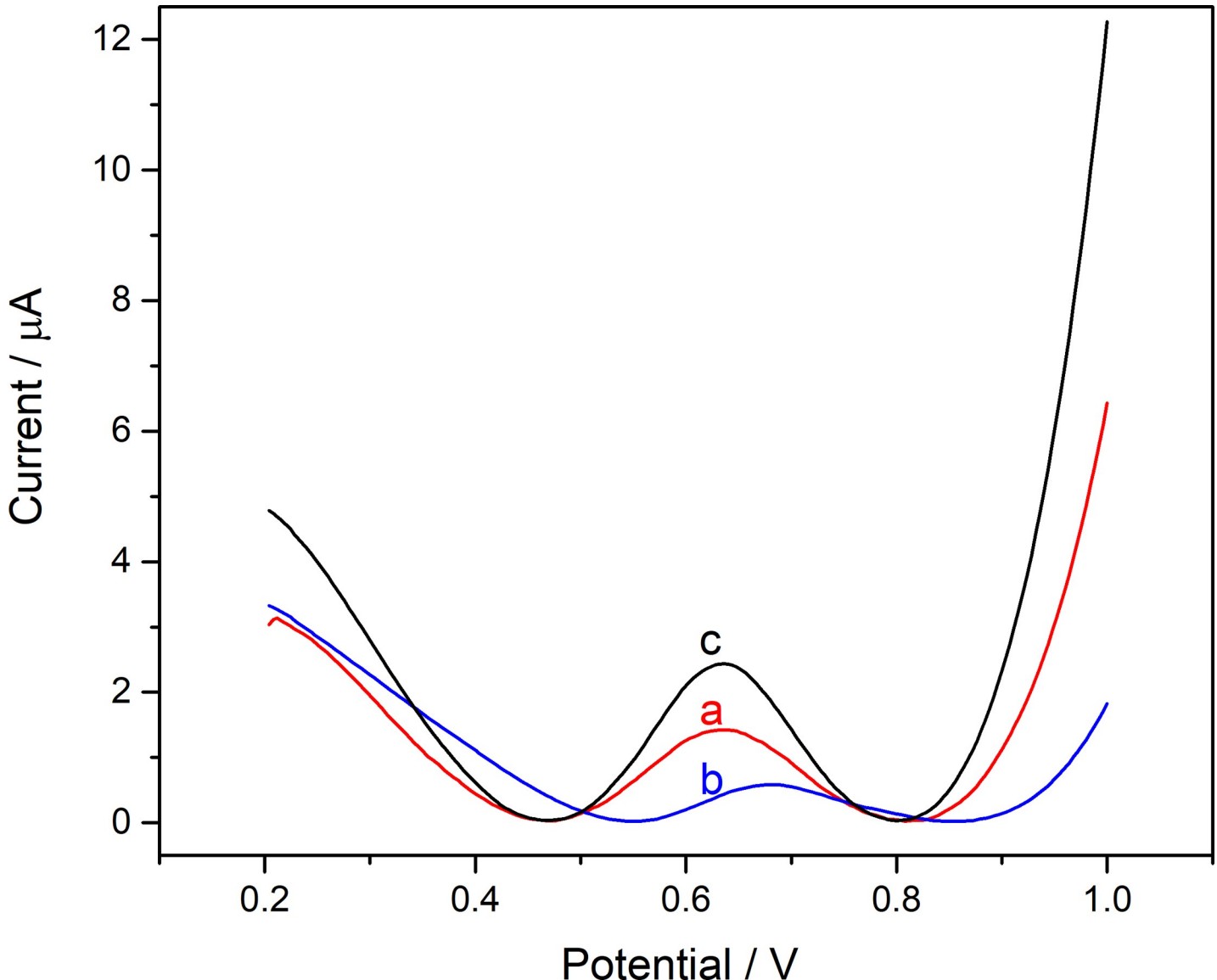

**Fig 7.** The DPVs of (a) AChE/CS/GR/GC, (b) AChE/Ti$_3$C$_2$T$_x$-CS/GC and (c) AChE/Ti$_3$C$_2$T$_x$-CS/GR/GCE in PBS containing 1 mM ATCl.

repeatability of the sensor. These sensors were tested DVP in 1 mM ATCl and showed very good repeatability, with RSD of peak current values for only 2.491% (S3 Fig). The fabricated biosensor can be stored at room temperature immersing in PBS. It is so stable that there was still 95% of the initial catalytic current value after 40 days (S4 Fig) [38, 39]. The PBS, NaCl, KCl, glucose and BSA solution was used to characterize the selectivity of the biosensor as shown in Fig 10B. In these solutions, the biosensor still had a high catalytic current, but in the DDVP solution, only about half of the catalytic current compared with other solutions shows good selectivity.

At present, there are few reported studies on the application of MXene in the detection of OPs as shown in Table 3. Zhou et al. and Jiang et al. from the same laboratory have achieved good results using Ti$_3$C$_2$T$_x$-based sensors to detect malathion. Song et al. designed a complex biosensor and achieved good results in detecting methamidophos. In this study, AChE

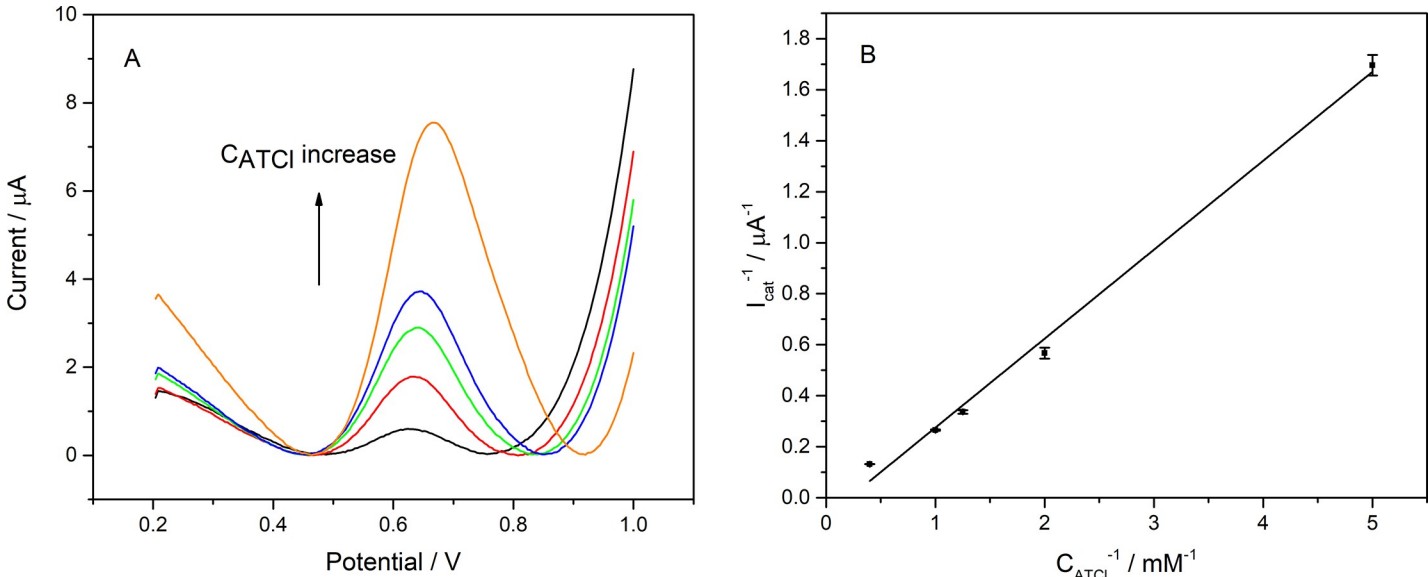

**Fig 8.** (A) The DPV responses of the AChE/Ti3C2Tx-CS/GR/GCE to various concentrations of ATCl and (B) the plot of 1/Icat versus 1/$C_{ATCl}$. n = 3.

biosensors with $Ti_3C_2T_x$ modification also showed better performance compared to those without $Ti_3C_2T_x$ modification. In addition, the biosensor can be stored at room temperature and exhibits excellent stability.

## Conclusions

Herein, An AChE electrochemical biosensor with a structure of AChE/$Ti_3C_2T_x$-CS/GR/GCE was prepared by layer-by-layer casting. The biosensor exhibits obviously electrochemical signal amplification with GR and $Ti_3C_2T_x$ modifying and increasing the effective area of the GCE. In addition, under the contribution of the large specific surface area of $Ti_3C_2T_x$, the

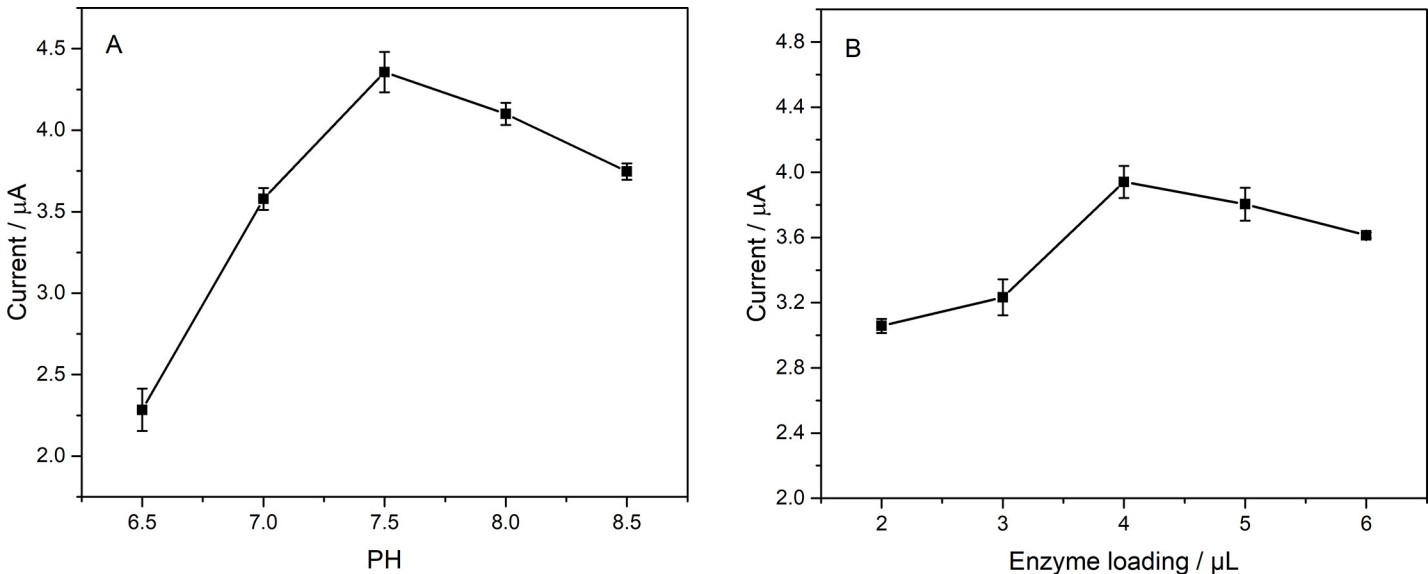

**Fig 9.** $I_{cat}$ of AChE/$Ti_3C_2T_x$-CS/GR/GCE in ATCl at (A) different pH and (B) enzyme loading. n = 3.

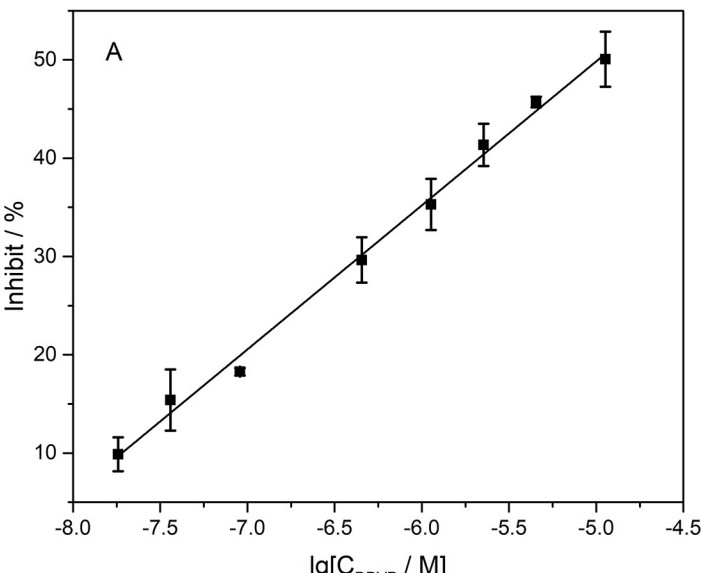
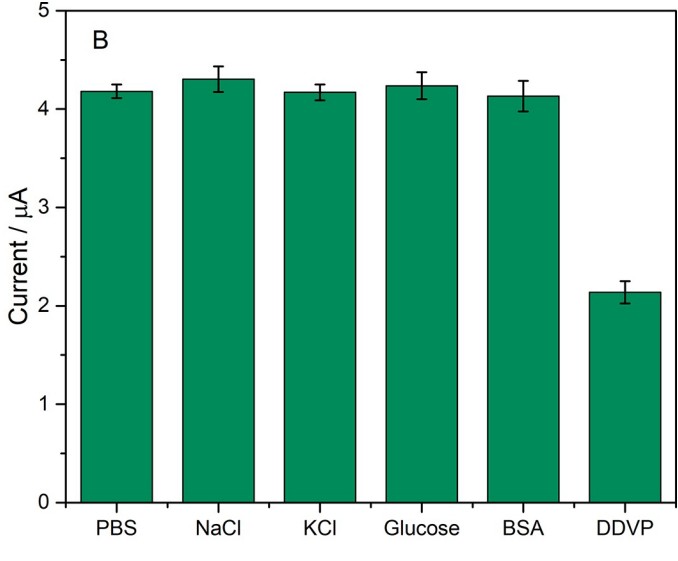

**Fig 10.** (A) The Inhibit% of the AChE/Ti$_3$C$_2$T$_x$-CS/GR/GCE biosensor versus the logarithm of DDVP concentration including: 18.1, 36.2, 91.0, 452.5 nM and 1.13, 2.26, 4.53, 11.31 μM. n = 3; (B) The DPV peak currents in 1mM ATCl of AChE/Ti$_3$C$_2$T$_x$-CS/GR/GCE biosensor after incubated in different substrates. The concentrations of NaCl, KCl, glucose, DDVP were 22.63 μM and BSA was 1 mg/ml. n = 3.

**Table 2. The DDVP recovery ratios of AChE/Ti$_3$C$_2$T$_x$-CS/GR/GCE biosensor in urban tap water samples.**

| Sample No. | DDVP Added (μM) | DDVP detected (μM) | Recovery (%) | RSD (%) (n = 3) |
|---|---|---|---|---|
| 1 | 10 | 9.82 | 98.2 | 9.435 |
| 2 | 1 | 1.09 | 109 | 5.517 |
| 3 | 0.1 | 0.99 | 99 | 6.701 |

**Table 3. Comparison with literature reported AChE biosensors.**

| Biosensor structure | Linear range / μM | LOD / nM | Analyte | References |
|---|---|---|---|---|
| AChE/Ti$_3$C$_2$T$_x$-CS/GR/GCE | 11.31 to 1.83×10$^{-2}$ | 14.45 | Dichlorvos | This work. |
| AChE/CS-Ti$_3$C$_2$T$_x$/GCE | 10$^{-2}$ to 10$^{-8}$ | 3×10$^{-4}$ | Malathion | Zhou et al. [40] |
| AChE/Ag@Ti$_3$C$_2$T$_x$/GCE | 10$^{-2}$ to 10$^{-8}$ | 3.27×10$^{-6}$ | Malathion | Jiang et al. [41] |
| AChE-Chit/MXene/Au NPs/MnO$_2$/Mn$_3$O$_4$/GCE | 1 to 10$^{-6}$ | 1.34×10$^{-4}$ | Methamidophos | Song et al. [42] |

biosensor has strong electrochemical catalytic performance, in which the apparent Michaelis–Menten constant K$_m$ is 4.89 mM. Under the fabrication process parameters optimized, the linear range of biosensors for DDVP detecting is from 18.1 nM to 11.31 μM and the LOD was 14.45 nM. In addition, biosensors exhibit good stability and can be stored for 40 days in a room temperature. The developed AChE biosensor shows high stability, reproducibility, sensitivity, accuracy and application potential in the process of real sample testing.

## Supporting information

**S1 Fig. XPS spectrum of Ti$_3$C$_2$T$_x$ nanosheets.** The overall atomic% of Ti 2p, C 1s, O 1s and F 1s are 22.22%, 29.87%, 19.19% and 28.72%.
(DOCX)

**S2 Fig.** XPS spectrum of (A) Ti$_3$AlC$_2$ nanosheets, (B) Ti 2p and (C) O 1s. The overall atomic% of Ti 2p, C 1s, O 1s, F 1s and Al 2p are 15.04%, 32.24%, 28.06%, 13.02% and 11.63%. Binding energy values of each bond associated with deconvoluted peaks are listed in S1 Table. (DOCX)

**S3 Fig. DPV results of five AChE/Ti$_3$C$_2$T$_x$-CS/GR/GCE biosensors which were prepared under same production parameters.** The DPV peak current of each biosensor is 4.210, 4.139, 4.271, 4.012 and 4.085 μA. And the RSD of the DPV results is 2.491%. (DOCX)

**S4 Fig. DPV results of AChE/Ti$_3$C$_2$T$_x$-CS/GR/GCE biosensor, which was stored for 0, 20, 30, and 40 days at room temperature in PBS solution.** After 0, 20, 30, and 40 days, the peak current of the DPV of the biosensor were 4.270, 4.092, 4.067 and 4.052 μA, respectively. (DOCX)

**S1 Table. XPS peak fitting results for crumpled Ti$_3$AlC$_2$.** (DOCX)

## Author Contributions

**Conceptualization:** Bo Wang.

**Data curation:** Bo Wang, Wenhao Shu, Lianqiao Yang.

**Formal analysis:** Bo Wang, Wenhao Shu.

**Funding acquisition:** Jianhua Zhang.

**Investigation:** Bo Wang, Wenhao Shu.

**Methodology:** Bo Wang, Yiru Li, Huaying Hu, Lianqiao Yang, Jianhua Zhang.

**Project administration:** Lianqiao Yang, Jianhua Zhang.

**Writing – original draft:** Bo Wang.

**Writing – review & editing:** Bo Wang, Yiru Li, Huaying Hu.

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
