## [Decision Letter · Decision Letter 0]

31 Dec 2019

PONE-D-19-31451

Acetylcholinesterase electrochemical biosensors with graphene-transition metal carbides nanocomposites modified for detection of organophosphate pesticides

PLOS ONE

Dear Dr. Yang,

Thank you for submitting your manuscript to PLOS ONE. We have received expert review of your manuscript. You will see that revision of your manuscript is advised, and suggestions are offered for improving the manuscript and its impact.  If you are prepared to undertake the work required, I would be pleased to consider the paper for publication in PLOS ONE. Therefore, we invite you to submit a revised version of the manuscript that addresses the points raised during the review process.

For your guidance, reviewers' comments are appended below.

We would appreciate receiving your revised manuscript by February 15, 2020. To enhance the reproducibility of your results, we recommend that if applicable you deposit your laboratory protocols in protocols.io, where a protocol can be assigned its own identifier (DOI) such that it can be cited independently in the future. For instructions see: http://journals.plos.org/plosone/s/submission-guidelines#loc-laboratory-protocols

We look forward to receiving your revised manuscript.

Kind regards,

Shabi Abbas Zaidi, Ph.D.

Academic Editor

PLOS ONE

Journal Requirements:

Additional Editor Comments:

In addition to reviewers' comments, please do address these specific comments while revising your manuscript;

The role of metal carbide should be made clearer in the modification of graphene-based biosensor.Is there any effect on oxidation state of Titanium? How about the stability of Ti3C2Tx in the modified sensor?In the XRD plot, X-axis should be changed to 2 theta (degree).Manuscript should be check for typo errors.

Reviewers' comments:

Reviewer's Responses to Questions

**Comments to the Author**

1. Is the manuscript technically sound, and do the data support the conclusions?

Reviewer #1: Yes

Reviewer #2: Yes

2. Has the statistical analysis been performed appropriately and rigorously? 

Reviewer #1: Yes

Reviewer #2: Yes

3. Have the authors made all data underlying the findings in their manuscript fully available?

Reviewer #1: Yes

Reviewer #2: Yes

4. Is the manuscript presented in an intelligible fashion and written in standard English?

Reviewer #1: Yes

Reviewer #2: No

5. Review Comments to the Author

Reviewer #1: Authors have applied GR, MXene, Chitosan and acetylcholinesterase modified biosensor for OP pesticides detection. The work seems interesting and very useful for application in real sample analysis. However, some points need to be addressed before its publication. I suggest minor revision for this article.

My comments are the following.

1. English needs to be improved throughout the manuscript.

2. A comparison table (updatedciting recent works) should be included in the manuscript discussing the electrochemical

performance of fabricated biosensor with other MXene based sensors and biosensors towards pesticides detection.

3. Introduction should be improved. Much more should be added about MXene based sensors and its properties.

4. What do the prominent peaks in XRD of MXene reveal?

5. What is the atomic percentage of elements in XPS analysis.

6. Why authors have chosen GR and MXene nanocomposite as GR is itself is very conducting material. What is the novelty

of this work as numerous works have been reported about pesticide detection using MXene and GR sensors.

7. Abstract and conclusion should be different and should not repeat the data. I suggest to discuss experimental results

only in abstract while major findings should be discussed in conclusion section.

8. Figures should be more clear and of equal size and dimensions.

9. Literature should be updated. pls include recent articles related to MXene sensing to increase the impact of material. For

eg. TrAC Trends in Analytical Chemistry, 105 (2018) 424-435, Biosensors Bioelectronics, 107 (2018) 69-75

10. How the stability of enzyme biosensor was justified.

11. have authors done selectivity analysis of biosensor in presence of different pesticides as it is an important parameter.

Reviewer #2: This paper by Wang et.al presents the fabrication of a biosensor based on graphene-transition metal carbides nanocomposite for detection of organophosphate pesticides which are important food biomarkers. The experimental work is systematic and reasonably organized. The paper may be accepted after subject to addressing of following comments;

1. There are numerous English grammar, and sentence misappropriation mistakes throughout the manuscript. For instance, the caption of Fig. 1C shall be replaced with Fig. 1D and vice versa. The word “purchased” should be included in the first line in “Materials and chemicals”.

2. Fig. 2B XRD exhibits very sharp peak of (103) and (105) which correspond to MAX phase. This indicates that there is considerable unetched MAX phase, unable to convert to MXene. Please comment in the light of reference 10.1016/j.jallcom.2018.04.152, Materials Science and Engineering B 191 (2015) 33–40 etc.

3. Line 154, page 11, please remove “and the”. Similarly correct the sentence in line 131 on page 11.

4. The EIS spectra curve (c) in Fig. 3B has different behavior as compared to other three curves in the higher frequency range (other have two semi-circles whereas curve (c) has a single semi-circle. Please comment.

5. Authors explain in Fig.5 that GR-modified biosensor is better in performance as compared to MXene-modified biosensor, however the synergistic enhancement occurs when biosensor is fabricated with MXene-GR-modification. What is the exact mechanism for enhancement of catalytic ability in composite biosensor?

6. What is the reason for high recovery (%) of 109 for sample 2 in Table 1.

6. PLOS authors have the option to publish the peer review history of their article (what does this mean?). If published, this will include your full peer review and any attached files.

Reviewer #1: No

Reviewer #2: No

---

## [Author Response · Author response to Decision Letter 0]

18 Mar 2020

Dear editor and reviewers,

Thank you very much for your email with which you sent us the reviewer’s report on our paper with reference number PONE-D-19-31451. We also wish to take this opportunity to thank the reviewer for the constructive comments and valuable recommendations. We have carefully revised the manuscript according to the reviewer’s suggestion. 

At last, please allow us to express our appreciation to your observations and suggestions for improvement of our article. I sincerely hope that the revised manuscript is now suitable for publication, if still not, one more chance to improve the quality of manuscript will be greatly appreciated.

Yours Sincerely,

Lianqiao Yang

Responses to Editor

Our responses to the comments and questions of editor are listed below and the corresponding revisions are marked in red in the revised manuscript.

Responses to Reviewer:1

Our responses to the comments and questions of editor are listed below and the corresponding revisions are marked in blue in the revised manuscript.

Comments: 

Authors have applied GR, MXene, Chitosan and acetylcholinesterase modified biosensor for OP pesticides detection. The work seems interesting and very useful for application in real sample analysis. However, some points need to be addressed before its publication.

Reply: Your recognition to our work and constructive suggestions are greatly appreciated.

Responses to Reviewer: 2

Our responses to the comments and questions of editor are listed below and the corresponding revisions are marked in green in the revised manuscript

Comments: 

This paper by Wang et.al presents the fabrication of a biosensor based on graphene-transition metal carbides nanocomposite for detection of organophosphate pesticides which are important food biomarkers. The experimental work is systematic and reasonably organized. The paper may be accepted after subject to addressing of following comments.

Reply: Dear reviewer, your recognition to our work and constructive suggestions are greatly appreciated.

For specific responses to questions from editors and reviewers, see the document of "response to reviewers.docx"

---

## [Decision Letter · Decision Letter 1]

6 Apr 2020

Acetylcholinesterase electrochemical biosensors with graphene-transition metal carbides nanocomposites modified for detection of organophosphate pesticides

PONE-D-19-31451R1

Dear Dr. Yang,

We are pleased to inform you that your manuscript has been judged scientifically suitable for publication and will be formally accepted for publication once it complies with all outstanding technical requirements.

With kind regards,

Shabi Abbas Zaidi, Ph.D.

Academic Editor

PLOS ONE

Additional Editor Comments (optional):

Reviewers' comments:

Reviewer's Responses to Questions

**Comments to the Author**

1. If the authors have adequately addressed your comments raised in a previous round of review and you feel that this manuscript is now acceptable for publication, you may indicate that here to bypass the “Comments to the Author” section, enter your conflict of interest statement in the “Confidential to Editor” section, and submit your "Accept" recommendation.

Reviewer #1: All comments have been addressed

Reviewer #2: All comments have been addressed

2. Is the manuscript technically sound, and do the data support the conclusions?

Reviewer #1: Yes

Reviewer #2: Yes

3. Has the statistical analysis been performed appropriately and rigorously? 

Reviewer #1: Yes

Reviewer #2: Yes

4. Have the authors made all data underlying the findings in their manuscript fully available?

Reviewer #1: Yes

Reviewer #2: Yes

5. Is the manuscript presented in an intelligible fashion and written in standard English?

Reviewer #1: Yes

Reviewer #2: Yes

6. Review Comments to the Author

Reviewer #1: Authors have well answered the queries raised and have described logically each comment in the revised version of manuscript. I recommend the manuscript to be published and accepted in its current form.

Reviewer #2: The authors have addressed all the comments. However, due to confusion, authors have inserted the word "Purchase" in the heading "Purchased materials and chemicals". Please remove this word "purchased" from heading and instead insert in the first line of the paragraph as "AChE (from electric eel), ATCl, and DDVP were purchased from Sigma-Aldrich".

7. PLOS authors have the option to publish the peer review history of their article (what does this mean?). If published, this will include your full peer review and any attached files.

Reviewer #1: No

Reviewer #2: Yes: Faisal Shahzad

---

## [Editor Report · Acceptance letter]

9 Apr 2020

PONE-D-19-31451R1 

Acetylcholinesterase electrochemical biosensors with graphene-transition metal carbides nanocomposites modified for detection of organophosphate pesticides 

Dear Dr. yang:

I am pleased to inform you that your manuscript has been deemed suitable for publication in PLOS ONE. Congratulations! Your manuscript is now with our production department. 

With kind regards,

on behalf of

Dr. Shabi Abbas Zaidi 

Academic Editor

PLOS ONE